# A guilt-free strategy increases self-reported non-compliance with COVID-19 preventive measures: Experimental evidence from 12 countries

**Jean-François Daoust**[1]*, **Éric Bélanger**[2], **Ruth Dassonneville**[3], **Erick Lachapelle**[3], **Richard Nadeau**[3], **Michael Becher**[4], **Sylvain Brouard**[5], **Martial Foucault**[5], **Christoph Hönnige**[6], **Daniel Stegmueller**[7]

**1** University of Edinburgh, Edinburgh, United Kingdom, **2** McGill University, Montreal, Canada, **3** Université de Montréal, Montreal, Canada, **4** IAST & IE University, Segovia, Spain, **5** Sciences Po–CEVIPOF, Paris, France, **6** University of Hanover, Hanover, Germany, **7** Duke University, Durham, North Carolina, United States of America

* jf.daoust@ed.ac.uk

**Data Availability Statement:** Replication files (data and code) are publicly accessible on Harvard Dataverse: Daoust, Jean-François; Éric Bélanger;

## Abstract

Studies of citizens' compliance with COVID-19 preventive measures routinely rely on survey data. While such data are essential, public health restrictions provide clear signals of what is socially desirable in this context, creating a potential source of response bias in self-reported measures of compliance. In this research, we examine whether the results of a guilt-free strategy recently proposed to lessen this constraint are generalizable across twelve countries, and whether the treatment effect varies across subgroups. Our findings show that the guilt-free strategy is a useful tool in every country included, increasing respondents' proclivity to report non-compliance by 9 to 16 percentage points. This effect holds for different subgroups based on gender, age and education. We conclude that the inclusion of this strategy should be the new standard for survey research that aims to provide crucial data on the current pandemic.

## Introduction

In the fight against the spread of the COVID-19 virus, research that aims to explain compliance with public health preventive measures is of utmost importance. The severity of virus activity is, in no small part, a function of citizens' behaviours [1, 2]. Therefore, much research has focused on understanding who complies, and what are the socio-demographic and attitudinal correlates of (non)compliance. Answering these questions is critical, helping governments and public health agencies to gather reliable data on compliance with preventive measures (e.g., social distancing, the use of face masks, etc.). A great deal of this data comes from survey research.

The need for high quality data on compliance with public health measures has led some researchers to investigate the reliability of survey data more closely. In particular, some studies have examined the possibility that public health restrictions in the pandemic context have

Ruth Dassonneville; Erick Lachapelle; Richard Nadeau; Michael Becher; Sylvain Brouard; Martial Foucault; Christoph Hönnige; Daniel Stegmueller, 2021, "Replication Data for: A Guilt-Free Strategy Increases Self-Reported Non-Compliance with COVID-19 Preventive Measures: Experimental Evidence from 12 Countries", https://doi.org/10.7910/DVN/YUC5R0, Harvard Dataverse, V1.

**Funding:** J.-F. Daoust acknowledges the financial support from SSPS Open Access (University of Edinburgh). M. Foucault and S. Brouard acknowledge the financial support from ANR – REPEAT grant (Special COVID-19), CNRS, Fondation de l'innovation politique, as well as regions Nouvelle-Aquitaine and Occitanie. Richard Nadeau and Éric Bélanger acknowledge the financial support from the Social Sciences and Humanities Research Council (SSHRC/CRSH). M. Becher gratefully acknowledges IAST funding from the ANR under the Investments for the Future ("Investissements d'Avenir") program, grant ANR-17-EURE-0010. D. Stegmueller acknowledges funding from Duke University and the National Research Foundation of Korea (NRF-2017S1A3A2066657). The funders had no role in study design, data collection and analysis, decision to publish, or preparation of the manuscript.

**Competing interests:** The authors have declared that no competing interests exist.

created common social norms for behaviours that are valued (e.g., social distancing). In turn, these norms create an incentive for respondents to under-report behaviours that are proscribed (e.g., social gatherings). The resulting social desirability bias can considerably affect the quality of data used by policy-makers and public health officials in their decision-making, which is problematic. Both Larsen et al. [3] and Munzert and Selb [4] considered the possibility that citizens' self-reported compliance with public health measures could be influenced by social desirability bias. These two single-country studies relied on a list experiment approach (also known as the unmatched item count technique). Reassuringly, they failed to detect a social desirability bias in the self-reported behaviour of Danish [3] and German [4] citizens. Findings from these list experiments, however, stand in contrast to the results of a recent study by Daoust et al. [5], which tested different "face-saving" (or guilt-free) strategies designed to loosen the social norm of compliance with public health measures in the context of three surveys conducted in Canada. The goal was to reduce social desirability in respondents' answers.

When exposed to a short preamble (referring to the fact that some people have altered their behaviours since the beginning of the pandemic, while others have continued to pursue various activities) combined with guilt-free answer choices (including "only when necessary"), respondents in the Daoust et al. [5] study were substantially more likely than respondents from the control group to report non-compliant behaviour in the context of the COVID-19 pandemic. This suggests that there is a social desirability bias in citizens' self-reported behaviour when no guilt-free option is provided. While promising, these results are based on a single country, i.e., Canada, and stand in contrast to the Danish and German studies cited above [3, 4]. As a result, we cannot be sure whether or not social desirability bias in reporting compliance with public health guidelines is particularly problematic in Canada, nor do we know how effective the guilt-free strategy is at reducing such bias beyond the Canadian case.

In this research, we extend the most effective face-saving strategy identified by Daoust et al. [5] to twelve countries. Doing so, we examine whether results are generalizable beyond Canada and across different contexts. In addition, we test whether the impact of the face-saving (guilt-free) answer option is homogenous across different subsets of the population to assess potential differential effects conditional on individual characteristics [6, 7]. To preview our results, we show that the guilt-free strategy is a very useful tool in every country examined, increasing respondents' proclivity to report non-compliance by 9 to 16 percentage points. This effect holds across gender, age and education subgroups. We conclude that this method should become the new standard for survey research of citizens' compliance with COVID-19 preventive measures, ultimately providing higher-quality data to governments and public health agencies.

## Measuring citizens' compliance with COVID-19 preventive measures

Social desirability bias has been a problem in survey research well before the COVID-19 pandemic and researchers have addressed the issue in several ways. Some work suggests that survey mode is an important factor, as the presence of a live interviewer can create greater incentives for respondents to provide more socially desirable answers. A meta-analysis on the topic suggests that an online mode is the best way to allow people to report undesirable behaviour [8]. But even with self-administered online surveys (i.e., without a live interviewer), respondents may still feel some incentive to provide what they deem to be socially desirable answers to survey questions. Survey researchers are therefore interested in developing additional ways and methodological tools to reduce social desirability and obtain more accurate estimates of undesirable behaviour [9].

Focusing on the current pandemic context, some of the first experimental studies of potential social desirability bias found limited evidence of this bias when employing the list experimental technique [3, 4]. Munzert and Selb [4] report a difference in the prevalence of not following social distancing between treated and untreated respondents of 6 percentage points that is "barely significant at conventional levels due to the vast measurement error of the list estimate." (p. 206) They also find that the prevalence of the undesirable behavior is higher when using a multi-valued response option compared to either a simple yes-no question or the list experimental estimate. This indicates that dichotomous direct questions should not be taken at face value. Discussing these findings, Munzert and Selb [4] suggest that face-saving strategies might be a "valuable alternative" (p. 207). In this study, we provide an experimental test of such a face-saving (or guilt-free) strategy across countries, and our results strengthen this conclusion.

The objective of the face-saving strategy is to reduce social desirability in respondents' answers by adding a (short) preamble and one or more guilt-free response options. We argue that such question and response wording can effectively loosen the norm around a desirable response and make it more acceptable for respondents to admit non-compliance with socially prescribed (and even mandated) behaviours. Such an approach has been applied to a wide range of topics. For example, political scientists have used it to study voter turnout [10, 11] where there is a clear norm that voting is the right thing to do. It has also proven to be useful to study illicit behavior like shoplifting [12], sexual/health behaviour such as using a condom during sexual intercourse [13, 14], or consumer choice in market research [15].

In the current context of the COVID-19 pandemic where it is crucial to understand citizens' level of compliance with preventive measures, Daoust et al. [5] applied the face-saving strategy to analyze the potential for a social desirability bias to affect survey responses obtained in Canada. This study provided evidence that face-saving strategies can increase the proportion of citizens who self-report non-compliance with a range of public health preventive measures in Canada. They argue that this increase in self-reported non-compliance is a consequence of reduced social desirability. While there is no objective benchmark to compare these estimates, they substantiate this claim by showing that a similar increase in self-reported non-compliance is not observed when the same face-saving strategy is applied to a series of placebo behaviours that are not prohibited (e.g., grocery shopping).

While promising, the results from Daoust et al. [5] suffer from a key limitation: Their focus on a single case, that is, Canada in April 2020. In this paper, we address this issue by implementing Daoust et al.'s [5] approach in twelve different countries. By doing so, we can ascertain whether results are specific to a single context and time period, or whether the effectiveness of a face-saving answer option to reduce social desirability in self-reported compliance with public health guidelines applies more generally. In the next section, we detail our data and how we implemented the face-saving strategy.

## Data and indicators

We ran a face-saving–also labeled as a 'guilt-free strategy' throughout the rest of the study– experiment in twelve countries: Australia, Austria, Brazil, France, Germany, Italy, New Zealand, Poland, Spain, Sweden, the United Kingdom and the United States. After outlining the study's details, we obtained respondents' written consent to participate in the survey. The data were analyzed anonymously, while ethical approval for the project "Citizens' Attitudes Under COVID-19 Pandemic" was received from TSE-IAST (Toulouse School of Economics). The online surveys were conducted by three different data collection firms: IPSOS (for all countries except Australia, United States and Spain), CSA (for Australia and the United States) and

Netquest (Spain). While these different platforms do not entail major differences, having different firms involved in data collection reduces the risk of bias due to potential "house effects." Data collection mainly occurred in mid-June 2020 within a period of a few days (maximum five), producing a nationally representative sample of about 1,000 respondents in most of the countries. In France, Germany and the USA, the sample size is about 2000 respondents. In these cases, half of the sample answered the questions used in this research.

At the time, countries in the study experienced different levels of infection and death rates [16] ranging from a low in Australia and New-Zealand (with less than .5 deaths per 100 000 inhabitants) to a high experienced in the United Kingdom (with more than 59 deaths per 100 000 inhabitants). Although the countries in our sample were similarly influenced by public health guidelines established by global health authorities, the timing, stringency and details of public health measures adopted to combat the pandemic also strongly varied across our cases, from Sweden (the least stringent) to New Zealand (the most stringent). Moreover, the countries in our sample also reflect different levels of issue politicization, with relatively more politicization of the pandemic response in Brazil and the USA relative to the other countries in our sample. Table A.1 of the S1 File lists the exact dates of data collection in each country, as well as the corresponding number of observations. Moreover, Table A.2 in S1 File shows the population per country and the death rates (per 100 000 inhabitants) as of June 15[th] while Table A.3 in S1 File provides an overview of the preventive measures that were recommended and compulsory in each country.

Here, we make use of the most effective face-saving strategy identified by Daoust et al. [5], which is also the strategy they recommend for future research (their Study 3). Extending this work to other countries, half of each national sample was randomly assigned to a direct question while the other half received the face-saving treatment. The direct question was: "Have you done any of the following activities in the last week?" followed by a set of four items and yes/no answer choices (skipping the question was possible but less than 0.5% did so). The face-saving question preamble was:

> "Some people have altered their behaviour since the beginning of the pandemic, while others have continued to pursue various activities. Some may also want to change their behaviour but cannot do so for different reasons. Have you done any of the following activities in the last week?"

Respondents in the treatment group received the answer options yes/occasionally/only when necessary/no. The first three answer choices indicated (and were coded as) non-compliance with the items. Of these three options, 'occasionally' and 'only when necessary' were the guilt-free answer choices. The four items, displayed in random order, were:

- Go shopping or take public transportation without a face mask or taking it off during it

- Meet friends, family or colleagues greeting them by shaking hands, hugging or kissing

- Have a group of friends or family over at your place

- Participate in social activities (work, sport, religious ceremony. . .) without respecting physical distancing

These items refer to behaviours that are crucial to minimizing the spread of the disease among the population, that is, wearing a face mask and practicing various forms of social distancing [17, 18]. Moreover, greeting people by shaking hands, hugging or kissing was clearly not recommended, while hosting a gathering at one's place was allowed though not without

some level of risk (e.g., Center for Disease Control and Prevention, see the "Hosting gatherings or cook-outs" section [19]).

In a second step, we explore whether the effects of the treatments are heterogeneous. We consider respondents' gender (female or male), age (treated as linear, from 18 to 91), and their level of education. Education was measured using different categories in each country given their different educational systems. We use a binary "university graduate" variable to model the effect of having obtained a university degree in each country. We also provide individual analyses for each country. Descriptive statistics summarizing the extent to which preventive measures were followed in each of the twelve countries can be found in Fig B.1 in S1 File, while descriptives for gender, age and education are reported in Appendix B of the S1 File. Moreover, Fig B.2 in the appendix of S1 File distinguishes between the two guilt-free answer choices ("Occasionaly" and "Only when necessary"), showing that there are no substantial differences between them with the only two exceptions being Brazil and Spain, where non-compliers tend to prefer the "only when necessary" answer choice.

## Results

Our main goal is to ascertain whether providing a short preamble and guilt-free answer options increases citizens' likelihood of reporting non-compliance with important public health measures like mask-wearing and social distancing. To shed light on this question, we analyze the data separately for the four items and the twelve countries. We thus provide a complete picture of the experimental effects and avoid pooling to ensure that the results are not driven by certain items or countries. Fig 1 displays the proportion of non-compliers with the preventive measures for the control (direct question) and treatment (face-saving) groups. The treatment group is depicted in light grey, while the dark grey bars indicate levels of self-reported compliance in the control condition. With four items in twelve countries, Fig 1 plots a total of 48 effects of interest.

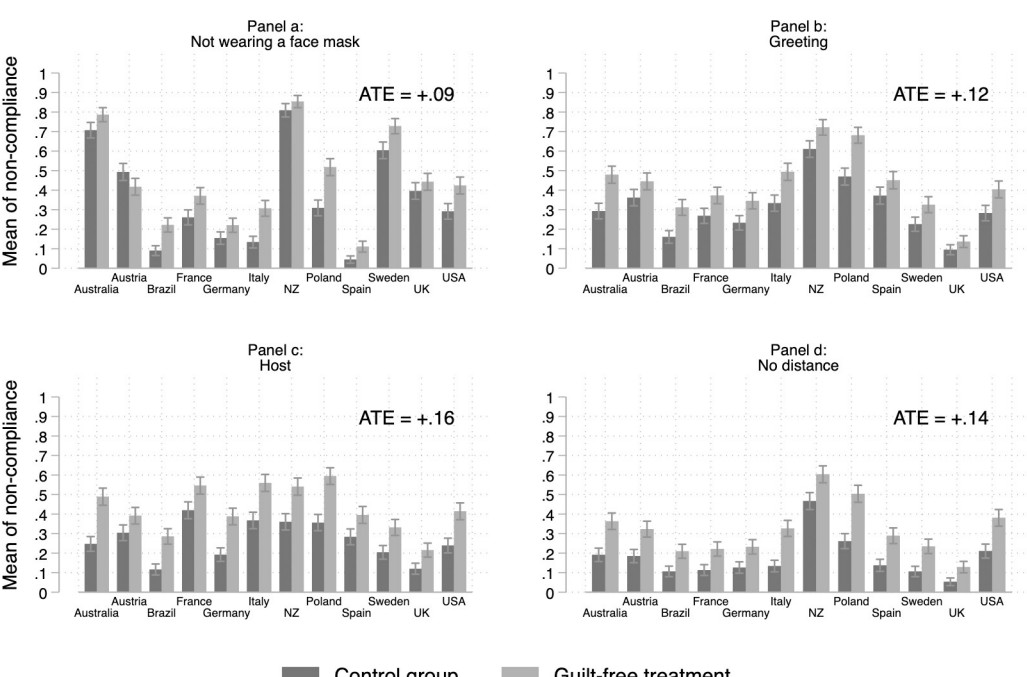

**Fig 1. Non-compliance in control and treatment (face-saving) groups.** Note: Means of non-compliance are shown with 95% confidence intervals included. ATE = Average treatment effect.

As is clear from eyeballing the graphs, the means of non-compliance are systematically higher in the face-saving group (the light grey bars). More precisely, the impact of the treatment is positive in 47 of 48 cases (the single exception being Austria for the face mask item). Substantially speaking, the average treatment effect (averaged across all countries) of receiving the treatment ranges from 9 (face mask) to 16 percentage points (hosting at home). Greeting people with non-recommended behaviours and not respecting physical distancing show an ATE of 12 and 14 percentage points, respectively. While we prefer to focus on the substantive effect, we note that in most cases (45 out of 47 positive effects), the differences are significant at $p < .05$ (based on two-sided $t$-test). Given the number of tests that we performed, we implemented the Romano-Wolf correction [20, 21]. Tables D.2-D.5 in S1 File show that the differences between the treatment and control groups are statistically significant (in 45 cases) even when using Romano-Wolf $p$-values.

Does the impact of the face-saving treatment differ across various subgroups of the population? In an exploratory fashion, we examine potential effect heterogeneity due to gender, age and education, which are known to be linked to compliance [5, 6, 22]. For this analysis, we rely on a pooled dataset that includes data from all countries. Fig 2 plots the coefficients from logistic regressions across different subgroups. The full regression outputs can be found in Tables C.1-C.3 of Appendix C in S1 File. The results are quite clear: Based on the evidence, we cannot reject the null hypothesis of no moderation effect. The interaction coefficients never reach statistical significance at $p < .05$. The direction of the effects is split for gender, three out of four coefficients are negative for age and three of four are positive for education.

All in all, we find that the face-saving strategy is effective. In total, 47 out of 48 effects are positive, and, their effects are substantive, ranging from 9 to 16 percentage points. Moreover, the impact of the treatment effect is fairly homogenous. As shown in Fig 2, it does not seem to be conditioned by individuals' gender, age or level of education.

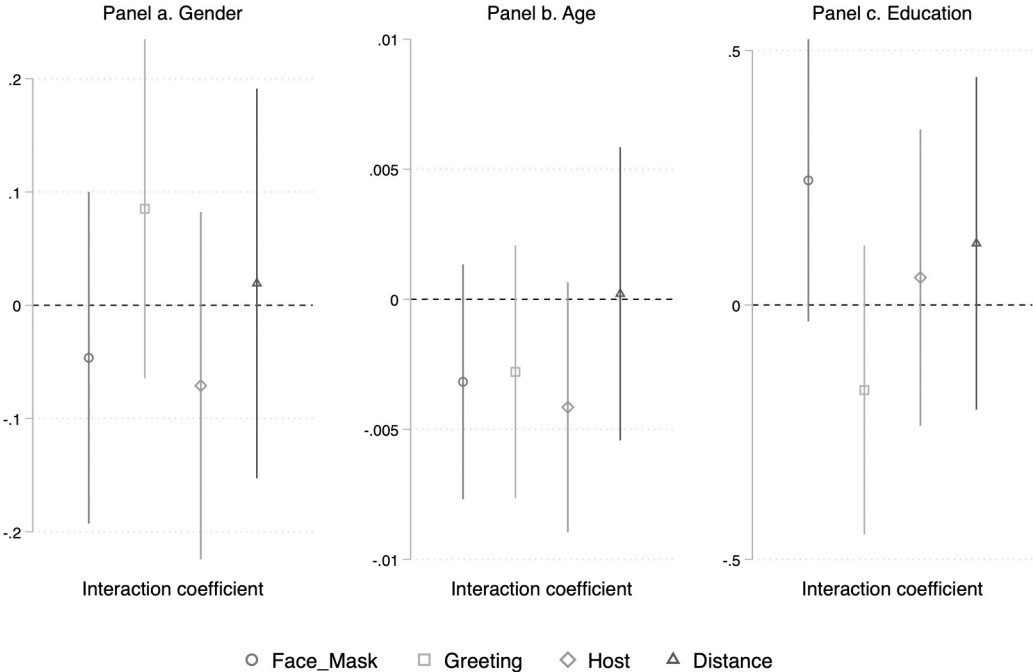

**Fig 2. Interaction coefficients for the guilt-free treatment, by gender, age and education.** Note: The values of the interaction coefficients estimated in Tables C.1-C.3 of the S1 File are shown, with 95% confidence intervals included.

We made sure that our conclusions were robust in several ways. First, although we should be cautious about randomization checks in an experimental context [23], we verified that both control and treatment groups were similar in terms of age, gender and education. The average age across treated and untreated respondents was identical (at 47 years of age); 52% of respondents in the control group were women compared to 51% in the treatment group, and means for the education variable were the same at .60.

Second, using weights for age, gender, education and region did not alter our conclusions. More specifically, we replicated Figs 1 and 2 for the weighted dataset (see Figs D.1 and D.2 in the S1 File). One interaction out of twelve reached $p < .05$, that is, the interaction between age and treatment for the 'no mask' behaviour. Focusing on the coefficients rather than $p$-values, our findings are very similar. Third, we made sure that results from our tests of heterogeneous treatment effects were not driven by unobserved characteristics of particular countries by estimating the effects with a model that included country fixed effects. Figs D.3 and D.4 in S1 File replicate and the results of this test and lead to essentially the same conclusion.

Fourth, we tackle the possibility that our results are in fact a "false positive" [24]. In a nutshell, our experimental design is based on the assumption that the differences in the proportion of self-reported non-compliance between the direct and face-saving questions are related to a reduced incentive to report socially desirable behaviours. While we think this is a very plausible assumption, we also examine the possibility that another mechanism might explain our results (for example the mere change in the number of response options) using a Canadian survey that was in field during the same period as the twelve surveys examined here. In this Canadian survey, we used the same face-saving strategy but with a broader battery of items, 8 instead of 4. This larger battery included behaviours that were not officially prohibited, i.e., where there should be much less social desirability. In panel A of Fig D.4 in S1 File, we show that there is a strong effect of the treatment (about 10 percentage points) for behaviours that the government prescribed, such as wearing a face mask in public, and that this effect is much less important in panel B (average of about 3 percentage points) for behaviours that perhaps entail a risk, but that the government did not proscribe–such as taking public transportation or shopping for non-essential products. While we do not have these 'placebo' items for twelve countries included in the analyses, this result for Canada is reassuring and increases our confidence that the greater proportion of self-reported non-compliance in the face-saving group is not a false positive.

Fifth, we also check whether or not a potential experimenter demand effect [25] might drive the results using data from a survey experiment in France. The design of the experiment is presented in the S1 File. Beyond both question formats analyzed earlier, we rely on a list experiment (and a question used by Munzert & Selb [4]) to estimate the levels of compliance with preventive behaviours from alternative measurement strategies. Results are shown in Fig D.5 of the S1 File. The list experiment approach was designed to reduce social desirability bias and is usually employed for this purpose [26]. Most importantly, the risk that there would be an experimenter demand effect is very limited. Overall, even if this experiment only includes one country, using estimates from the list experiment as a baseline brings no evidence of an experimenter demand effect associated to the guilt-free question format that would systematically bias our estimates.

## Discussion and implications

West et al. [27: 451] argued that "there is an urgent need to develop and evaluate interventions to promote effective enactment of these behaviours and provide a preliminary analysis to help guide this." We agree with the authors and address the issue of social desirability and its impact on self-reported compliance with COVID-19 public health guidelines. While the work of Daoust et al. [5]

on this topic showed a face-saving strategy is a promising approach to attenuate social desirability, evidence was lacking on whether this face-saving strategy was effective beyond Canada.

In this research, we tested the face-saving strategy using a survey experiment in twelve countries to examine the benefits of this approach. We replicated the findings found in Daoust et al. [5] and most importantly, did so in a diverse set of contexts, with different countries and political systems, in different stages of the pandemic (deconfinement in most countries), with different levels of infections, etc. Based on four public health preventive measures related to the wearing of face masks, greetings, hosting gatherings at one's home, and social distancing, we found that the face-saving strategy increased the proportion of citizens who readily answer that they did not comply in 47 out of 48 cases. Most importantly, the effects were substantial. They ranged from 9 percentage points (the mask item) to 16 percentage points (hosting at home) and are robust to several additional tests. Given that there are no readily available, objective benchmarks for the four preventive measures analyzed in our study, we acknowledge that there is no way to compare our estimates with external measures. However, our experimental design and extensive robustness checks bolster the conclusion that the differences we find can be at least partly attributed to a social desirability bias.

There have already been major advances in the development of observational measures of citizens' behaviour (i.e., not from survey data). Among others, France has recently used cameras in its subway stations to quantify the proportion of people who wear a mask when using public transportation, and several countries are developing applications to track the inter-regional movement of their residents [28]. While useful, we cannot rely solely on behavioral data in the fight against the pandemic. First, observational measures are not available for several important preventive public health measures, as many measures cannot be examined in public, such as the respect of social distancing if one hosts a gathering at their private home. Second, even behavioral data like that obtained from cameras or tracing applications have some major drawbacks. Most importantly, due to technical and privacy constraints, this approach does not easily provide researchers important auxilliary information about who complies and what makes people comply or not. For these reasons, we believe that survey research remains a crucial complement to other data sources.

In summary, policymakers and public health experts require survey data, and as a result, we should aim for data that is of the best possible quality. Our research confirms that using a guilt-free strategy is an effective approach and is relevant to anyone who aims to provide data on citizens' compliance with COVID-19 preventive measures. This type of data is crucial for governments and public health agencies to make enlightened decisions. Moreover, as the strategy simply implies the addition of a very short preamble and guilt-free answer choices, there are very limited additional costs involved to implement this method compared to a direct question. While replications would be welcome to strengthen the validity of the approach, we believe that our comparative research provides a firm ground for what should become the standard when measuring citizens' compliance with public health preventive measures.

## Supporting information

**S1 File.**
(PDF)

## Acknowledgments

We thank all everyone involved in the data collection process, including: Pavlos Vasilopoulos (University of York), Vincenzo Galasso (Bocconi University), Eric Kerrouche (Sciences Po–

CEVIPOF), Sandra Leon Alfonso (University Carlos III), Vincent Pons (Harvard Business School), Hanspeter Kriesi (EUI), Dominique Reynié (Sciences Po–CEVIPOF). We also thank Eric Kennedy (York University) who provided the opportunity to present a first version of this research as part of the COVID-19 Survey Research Working Group seminars, and we are grateful for participants' comments. Any error is ours.

## Author Contributions

**Conceptualization:** Jean-François Daoust.

**Formal analysis:** Jean-François Daoust.

**Funding acquisition:** Éric Bélanger, Ruth Dassonneville, Erick Lachapelle, Richard Nadeau, Michael Becher, Sylvain Brouard, Martial Foucault, Christoph Hönnige, Daniel Stegmueller.

**Visualization:** Jean-François Daoust.

**Writing – original draft:** Jean-François Daoust, Éric Bélanger, Ruth Dassonneville, Erick Lachapelle, Richard Nadeau, Michael Becher, Sylvain Brouard, Martial Foucault, Christoph Hönnige, Daniel Stegmueller.

**Writing – review & editing:** Jean-François Daoust, Éric Bélanger, Ruth Dassonneville, Erick Lachapelle, Richard Nadeau, Michael Becher, Sylvain Brouard, Martial Foucault, Christoph Hönnige, Daniel Stegmueller.

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
