## [Decision Letter · Decision Letter 0]

21 Dec 2020

PONE-D-20-32568

Face-Saving Strategies Increase Self-Reported Non-Compliance with COVID-19 Preventive Measures: Experimental Evidence from 12 Countries

PLOS ONE

Dear Dr. Daoust,

Thank you for submitting your manuscript to PLOS ONE. After careful consideration, we feel that it has merit but does not fully meet PLOS ONE’s publication criteria as it currently stands. Therefore, we invite you to submit a revised version of the manuscript that addresses the points raised during the review process.

The three reviewers agree that the paper is well written and addresses a relevant and timely question.

However, all of them raise a number of concerns and provide few suggestions that, if properly addressed, will help to further improve the quality of the paper.

More specifically, I agree with Reviewer 1 that you should discuss more in detail how the different number of possible answers in the questions posed to the treatment and the control group may influence the main results.

Similarly, Reviewer 3 raises some concerns on the scale used for potential answers, arguing that this empirical strategy may not apply to behaviors whose sensitivity is not "gradual" in nature.

In the same vein, Reviewer 2 asks to tackle more directly the issue of “experimenter demand effect”, providing a number of useful references.

We look forward to receiving your revised manuscript.

Kind regards,

Federica Maria Origo

Academic Editor

PLOS ONE

Journal Requirements:

2.Please provide additional details regarding participant consent. In the ethics statement in the Methods and online submission information, please ensure that you have specified (1) whether consent was informed and (2) what type you obtained (for instance, written or verbal, and if verbal, how it was documented and witnessed). If your study included minors, state whether you obtained consent from parents or guardians. If the need for consent was waived by the ethics committee, please include this information.

3.In your Data Availability statement, you have not specified where the minimal data set underlying the results described in your manuscript can be found. PLOS defines a study's minimal data set as the underlying data used to reach the conclusions drawn in the manuscript and any additional data required to replicate the reported study findings in their entirety. All PLOS journals require that the minimal data set be made fully available. For more information about our data policy, please see http://journals.plos.org/plosone/s/data-availability.

Reviewers' comments:

Reviewer's Responses to Questions

**Comments to the Author**

1. Is the manuscript technically sound, and do the data support the conclusions?

Reviewer #1: Partly

Reviewer #2: Yes

Reviewer #3: Yes

2. Has the statistical analysis been performed appropriately and rigorously? 

Reviewer #1: Yes

Reviewer #2: Yes

Reviewer #3: Yes

3. Have the authors made all data underlying the findings in their manuscript fully available?

Reviewer #1: Yes

Reviewer #2: Yes

Reviewer #3: Yes

4. Is the manuscript presented in an intelligible fashion and written in standard English?

Reviewer #1: Yes

Reviewer #2: Yes

Reviewer #3: Yes

5. Review Comments to the Author

Reviewer #1: The paper is written in a clear way and addresses an important question. Here are some comments:

- The authors write (page 12), "among several important subgroups of the population including gender, age and education". This implies that subgroups other than gender, age and education have been checked, but there is no evidence of that in the paper. Please, clarify.

- The authors write (page 11), "The results for potential heterogeneous effects across subgroups based on gender, age and education are quite clear: overall there is no substantial moderation effect. The interaction coefficients never reach statistical significance at p<.05." Then, at page 12, they write, referring to the weighted dataset "One interaction reaches p<.05 for age but the single significant interaction for education now fails to pass that threshold". The two sentences appear to be in contradiction with each other.

- In the discussion, the authors should go back to the results by Munzert and Selb (2020) on Germany and discuss why the findings differ.

- My main concern about the methodology adopted by the authors is that they compare a treatment with 2 options (yes/no) to a face saving treatment with 4 options (yes/occasionally/only when necessary/no), where 3 (yes/occasionally/only when necessary) correspond to the single "yes" option in the direct question. The issue is that, if a subset of the sample answers randomly, then this will inflate the positive answers in the face saving treatment, as there is a 75% probability that a random answer is "yes/occasionally/only when necessary" vs a 50% probability that a random answer is "yes" in the direct question. This would generate spurious evidence of social desirability bias. The authors should address this concern.

Reviewer #2: This paper tackles the important issue of correcting for social-desirability when eliciting self-reported preventive measures to prevent the spread of COVID-19.

The paper is clearly written, concise, well-executed. The data collection effort across 12 countries is impressive, and well harmonized, providing interesting comparisons in both average reported behaviours and the response to the “treatment”. The topic is certainly relevant for public policy and very current.

I have one major concern—the fact that the “face-saving” treatment might correct for social desirability but could introduce experimenter demand effects—a minor concern—related to inference—and a few minor comments and suggestions, which I outline below in bullet points.

*** Major Concern***

(1) Experimenter demand effect

The “face-saving” treatment frames the question in such a way that people feel OK reporting a lower level of preventive behaviour. This framing might have two effects: (1) contrast social desirability upward-bias; and (2) introduce an experimenter demand effect downward-bias. The intent of the author is to tackle the first bias, but I am afraid that they are opening the door to the second type of bias. Since there is no objective external report of these behaviours, there is no benchmark to understand how much this “downward treatment effect” is actually correcting for the first upward bias, or going too far and introducing the second downward bias.

Would it be possible to have a (more) objective or external measure of the actual level of (some of) these preventive behaviours? If not for the individuals in the survey, at least on average for these countries? For example, some mobility data collected by google https://www.google.com/covid19/mobility/ or apple https://covid19.apple.com/mobility

Alternatively, recent papers have tried to tackle the issue of experimenter demand effect (see for example de Quidt et al. 2018 and the citations therein): would it be possible to run a small pilot study in one of these countries to show that experimenter demand effect is not an issue?

If none of these alternatives are feasible, I would at least suggest discussing this issue in the paper.

Regardless of whether this additional analysis is performed, I strongly urge the authors to mention in the paper that there is no way of knowing the “real” level of preventing behaviours, and that all of the statistics that are shown in the paper come from self-reports which cannot be validated.

*** Minor Concern***

(2) Confidence Interval

I believe that reporting 84% confidence intervals is misleading. I understand that Macgregor-Fors & Payton 2013 suggest it for “visual inspections” of overlapping confidence intervals, but one should NOT consider overlapping CI as a test of difference between plotted coefficients.

I would strongly suggest reporting always and only 99% and/or 95% confidence intervals.

Also, given the number of countries and behaviours reported, it would be nice to have a test correcting for multiple hypothesis testing, in the vein of Romano and Wolf (2005, 2017)---for example, a stata command that implements it is available here https://ideas.repec.org/c/boc/bocode/s458276.html and an R command here https://rdrr.io/github/grayclhn/oosanalysis-R-library/man/stepm.html

*** Questions / Comments / Suggestions ***

• I find the terminology “face-saving strategies” a bit confusing in the setting of the COVID-19 pandemic: when I first read the title, I thought that the authors were referring to wearing a mask (I confused it with “face-covering”) or some other form of preventive behaviour (which could be “saving” lives). I would suggest something like “plausible deniability” or “guilt-free”. However, this is a purely personal comment, and need not be taken into consideration if the authors do not agree.

• For the figures, I would suggest ordering the countries from highest to lowest reported behaviour, and maybe plot them vertically. See for example Cohn et al. (2019).

• The graphs and tables have only minimal notes, it would be useful for the cursory reader to have more information available right at the bottom of the graph

• It could be helpful to have a table or a list of the behaviours that are allowed/suggested/prohibited in the different countries at the time of the data collection, or some form of stringency of the measures used by each country, similar to https://www.bsg.ox.ac.uk/research/research-projects/coronavirus-government-response-tracker#data or https://ourworldindata.org/grapher/covid-stringency-index

• The precise definition of the treatment comes only at the bottom of page 7. It might be useful for the reader to have it earlier, maybe even in the introduction.

References:

Clarke, D., Romano, J. P., & Wolf, M. (2020). The Romano-Wolf Multiple Hypothesis Correction in Stata. IZA Working Paper. https://www.iza.org/publications/dp/12845/the-romano-wolf-multiple-hypothesis-correction-in-stata

Romano, J. P., & Wolf, M. (2017). Resurrecting weighted least squares. Journal of Econometrics, 197(1), 1–19. https://doi.org/10.1016/j.jeconom.2016.10.003

Romano, J. P., & Wolf, M. (2005). Exact and Approximate Stepdown Methods for Multiple Hy-pothesis Testing. Journal of the American Statistical Association, 100(469), 94–108. https://doi.org/10.1198/016214504000000539

Cohn, A., Marechal, M. A., Tannenbaum, D., & Zünd, C. L. (2019). Civic honesty around the globe. Science, eaau8712. https://doi.org/10.1126/science.aau8712

de Quidt, J., Haushofer, J., & Roth, C. P. (2018). Measuring and Bounding Experimenter Demand. American Economic Review, 108(11), 3266–3302. https://doi.org/10.1257/aer.20171330

Reviewer #3: A very nice paper, well written, very transparent. My only concern is, that their claim of using a "new strategy" is a bit overstretched. Below I give more details on that criticism. Still I am convinced that the paper is a valuable contribution to the sensitive question literature and - with its focus on surveying adherence to covid preventive measures - is highly relevant.

• What are the main claims of the paper and how significant are they for the discipline?

• The authors replicate a previous study about the effect of face-saving strategies on self-reports of non-compliance when surveying compliance to covid-19 measures (e.g. social distancing). They ran the experiment in twelve countries and find consistent evidence of a positive effect of the strategy under investigation. The topic is highly relevant because it tackles a survey methodology issue that is of utmost importance under the actualy circumstances (the pandemic).

• Are the claims properly placed in the context of the previous literature? Have the authors treated the literature fairly?

• In general yes. My only criticism is, that what they present as „new face-saving strategy“ is basically simply choosing likert-scale response categories instead of dichotomous yes/no response option. In addition, only providing the yes or no response options are for most of the surveyed behaviors such as keeping distance, wearing masks, obviously not very suitable. I would guess, that not much researcher have used "primitive" yes/no response options for surveying adherence to such measures.

• Still, I find it valuable to have a close look at this topic and to study how response options influence answer behavior. But the authors should explain a bit more detailed in what regard their strategy is really new (or not so much).

• I also wonder, whether the authors could provide information about the effect of the response option and the question preamble separately. In their design these two aspects are combined.

• Another issue is, that their strategy is only applicable to sensitive behavours whose sensitivity is somehow "gradual" in nature. E.g. it works for shaking hands or wearing face masks. But it wouldn't work for having been arrested, committing tax fraud, or voting for a right-wing party. I would wish, authors would comment on that.

• Do the data and analyses fully support the claims? If not, what other evidence is required?

• Yes. Design and analysis are reported very carefully and transparently. Authors provide varous robustness checks, analysis by countries, additional details, and consider, for instance, the threat of „false positives“ that might invalidate their findings.

• Are original data deposited in appropriate repositories and accession/version numbers provided for genes, proteins, mutants, diseases, etc.?

• Yes

• Are details of the methodology sufficient to allow the experiments to be reproduced?

• Yes

• Is the manuscript well organized and written clearly enough to be accessible to non-specialists?

• Yes, very clearly written and organized. Nice to read! Presentation of results is exemplary, with very nice figures and corresponding tables in the appendix.

6. PLOS authors have the option to publish the peer review history of their article (what does this mean?). If published, this will include your full peer review and any attached files.

Reviewer #1: No

Reviewer #2: No

Reviewer #3: **Yes: **Marc Höglinger

---

## [Author Response · Author response to Decision Letter 0]

11 Feb 2021

Our responses include tables and figures so we uploaded a PDF document where we detail our answers. See the "Response to Reviewers" document.

---

## [Decision Letter · Decision Letter 1]

29 Mar 2021

A Guilt-Free Strategy to Increase Self-Reported Non-Compliance with COVID-19 Preventive Measures: Experimental Evidence from 12 Countries

PONE-D-20-32568R1

Dear Dr. Daoust,

We’re pleased to inform you that your manuscript has been judged scientifically suitable for publication and will be formally accepted for publication once it meets all outstanding technical requirements.

Kind regards,

Federica Maria Origo

Academic Editor

PLOS ONE

Additional Editor Comments (optional):

Reviewers' comments:

Reviewer's Responses to Questions

**Comments to the Author**

1. If the authors have adequately addressed your comments raised in a previous round of review and you feel that this manuscript is now acceptable for publication, you may indicate that here to bypass the “Comments to the Author” section, enter your conflict of interest statement in the “Confidential to Editor” section, and submit your "Accept" recommendation.

Reviewer #1: All comments have been addressed

Reviewer #2: All comments have been addressed

Reviewer #3: All comments have been addressed

2. Is the manuscript technically sound, and do the data support the conclusions?

Reviewer #1: Yes

Reviewer #2: Yes

Reviewer #3: Yes

3. Has the statistical analysis been performed appropriately and rigorously? 

Reviewer #1: Yes

Reviewer #2: Yes

Reviewer #3: Yes

4. Have the authors made all data underlying the findings in their manuscript fully available?

Reviewer #1: Yes

Reviewer #2: Yes

Reviewer #3: Yes

5. Is the manuscript presented in an intelligible fashion and written in standard English?

Reviewer #1: Yes

Reviewer #2: Yes

Reviewer #3: Yes

6. Review Comments to the Author

Reviewer #1: I really liked the revised version of the paper. Well done. It is a relevant contribution on an important topic.

Reviewer #2: Thank you for your through revisions. I very much appreciated the new findings from the list experimenter in France. Although the results for "friends" are a bit puzzling, I agree that the other results do not suggest any strong presence of an experimenter demand effect.

Reviewer #3: Authors have carefully considered all issues raised by the reviewers and have implemented the corresponding changes.

7. PLOS authors have the option to publish the peer review history of their article (what does this mean?). If published, this will include your full peer review and any attached files.

Reviewer #1: No

Reviewer #2: **Yes: **Pietro Biroli

Reviewer #3: **Yes: **Marc Höglinger

---

## [Editor Report · Acceptance letter]

8 Apr 2021

PONE-D-20-32568R1 

A Guilt-Free Strategy Increases Self-Reported Non-Compliance with COVID-19 Preventive Measures: Experimental Evidence from 12 Countries 

Dear Dr. Daoust:

I'm pleased to inform you that your manuscript has been deemed suitable for publication in PLOS ONE. Congratulations! Your manuscript is now with our production department. 

Kind regards, 

on behalf of

Dr. Federica Maria Origo 

Academic Editor

PLOS ONE